# Tumor-Educated Platelets in Urological Tumors: A Novel Biosource in Liquid Biopsy

**DOI:** 10.3390/ijms26083595

**Published:** 2025-04-11

**Authors:** Mariona Figols, Sviatoslav Chekhun, Maria Fernández-Saorin, Ignacio Pérez-Criado, Ana Bautista, Albert Font, Vicenç Ruiz de Porras

**Affiliations:** 1Medical Oncology Department, Althaia Xarxa Assistencial Universitària de Manresa, C/ Dr. Joan Soler, 1-3, 08243 Manresa, Spain; mfigols@althaia.cat (M.F.); iperezc@althaia.cat (I.P.-C.); ambautista@althaia.cat (A.B.); 2PhD Programme in Medicine and Biomedical Sciences, Doctoral School, University of Vic, Central University of Catalonia (UVic-UCC), C/ Dr. Junyent, 1, 08500 Vic, Spain; 3Faculty of Medicine, University of Vic, Central University of Catalonia (UVicUCC), Can Baumann, Ctra, de Roda, 70, 08500 Vic, Spain; 4CARE Program, Germans Trias i Pujol Research Institute (IGTP), Camí de les Escoles, s/n, 08916 Badalona, Spain; schekhun.germanstrias@gencat.cat (S.C.); mfernandezs@igtp.cat (M.F.-S.); afont@iconcologia.net (A.F.); 5Badalona Applied Research Group in Oncology (B⋅ARGO), Catalan Institute of Oncology, Camí de les Escoles, s/n, 08916 Badalona, Spain; 6Medical Oncology Department, Catalan Institute of Oncology, Camí de les Escoles, s/n, 08916 Badalona, Spain; 7GRET and Toxicology Unit, Department of Pharmacology, Toxicology and Therapeutic Chemistry, Faculty of Pharmacy and Food Sciences, University of Barcelona, 08028 Barcelona, Spain

**Keywords:** tumor-educated platelets, liquid biopsy, biomarkers, urological tumors

## Abstract

Platelets, traditionally recognized for their role in hemostasis, have emerged as pivotal players in cancer biology. They actively contribute to tumor proliferation, angiogenesis, immune evasion, and metastasis and thus play a significant role in cancer progression. Tumor-educated platelets (TEPs) acquire protumorigenic phenotypes through RNA, protein, and receptor profile alterations driven by interactions with tumors and their microenvironment. These modifications enable TEPs to enhance tumor growth and dissemination and to play a critical role throughout the metastatic process. Moreover, TEPs are promising biomarkers that can easily be analyzed in liquid biopsies. Since they dynamically mirror tumor activity through transcriptomic and proteomic changes, their analysis offers a non-invasive method for determining cancer detection and diagnosis, patient prognosis, therapy monitoring, and personalization of treatment. Their demonstrated accuracy in identifying cancer types and predicting treatment responses underscores their ability to provide real-time insights into tumor biology, including in urological malignancies. Their diagnostic potential and their accessibility as blood-sourced biomarkers position TEPs as transformative tools in advancing personalized oncology. Here, we focus on the role of TEPs in urological tumors, exploring their applications in early cancer detection, disease monitoring, and the design of tailored therapeutic strategies.

## 1. The Role of Platelets in Cancer: An Overview

Cancer is a leading cause of mortality worldwide, with its progression involving complex interactions between tumor cells and various components of the host’s circulatory and immune systems. Among these components, platelets have emerged as critical players in cancer biology [1].

Platelets, the smallest anucleated elements of the blood system, originate from megakaryocytes. While their role has traditionally been observed mainly in hemostasis and coagulation, they are now recognized as dynamic participants in various physiological and pathological processes, including inflammation and immune response [2]. They possess a rich array of surface receptors, and when activated, they release numerous bioactive molecules stored in their alpha-granules and dense granules. These molecules include growth factors, cytokines, chemokines, and adhesion molecules, which can modulate the behavior of other cells, including tumor cells. The interaction between tumors and platelets significantly influences various aspects of cancer development, including tumor proliferation, angiogenesis, immune evasion, and metastasis [3,4] (Figure 1). The subsequent sections present a broad review of the multiple facets of platelets in cancer development.

### 1.1. Platelets in Tumor Growth and Progression

Platelets are actively involved in promoting tumor growth and progression through various mechanisms. One of the key ways that they facilitate tumor growth is by secreting a wide range of growth factors, cytokines, and chemokines, which create a favorable microenvironment for tumor cells [5,6].

Platelet–tumor cell interactions can occur directly, facilitated by various adhesion molecules such as P-selectin, glycoprotein IIb/IIIa (GPIIb/IIIa), integrins, and CD40 ligand (CD40L). These interactions are crucial for the survival and dissemination of tumor cells. For example, P-selectin on platelets binds to its ligand on tumor cells, facilitating tumor cell arrest on the vascular endothelium, which is a critical step in tumor progression and metastasis [7].

Enhanced platelet activation is evident in patients with bladder cancer, as demonstrated by increased surface P-selectin expression and elevated levels of its soluble form. This activation plays a significant role in tumor growth and progression. A research team from Cairo University and the Theodor Bilharz Research Institute in Egypt evaluated 30 bladder cancer patients using flow cytometry and ELISA to measure these markers. Their findings suggest that P-selectin-mediated interactions contribute critically to tumor progression, highlighting this axis as a promising target for therapeutic intervention [8]

Platelets store and release a variety of growth factors, including platelet-derived growth factor (PDGF), transforming growth factor-beta (TGF-β), vascular endothelial growth factor (VEGF), and epidermal growth factor (EGF), all of which play crucial roles in cell proliferation, survival, and migration. In particular, PDGF and TGF-β enhance tumor cell proliferation and survival by activating signaling pathways that promote cell cycle progression and inhibit apoptosis [9]. The release of these growth factors by platelets can be triggered by various stimuli, including direct interaction with tumor cells, exposure to tumor-derived microparticles, and activation by thrombin or collagen [10].

### 1.2. Platelets in Tumor Angiogenesis

Angiogenesis, the growth of new blood vessels from existing structures, is a key factor in cancer, as these new vessels supply oxygen and nutrients that promote tumor growth and metastasis [11]. Platelets play a crucial role in angiogenesis by serving as reservoirs and active mediators of pro-angiogenic factors, thereby influencing the tumor microenvironment (TME). Platelets are rich in angiogenic molecules, such as VEGF, EGF, basic fibroblast growth factor (bFGF), insulin growth factors, and angiopoietins. Upon activation, platelets release these factors, which promote endothelial cell proliferation, migration, and tube formation—fundamental processes in the formation of new blood vessels. This dynamic contribution underscores the central role of platelets in driving angiogenesis, particularly in the context of tumor progression [12,13,14].

Renal cell carcinoma (RCC) is a paradigmatic tumor in its reliance on angiogenesis. In this context, platelets perform specialized functions that extend beyond their conventional pro-angiogenic role. They not only deliver VEGF to stimulate endothelial proliferation and tube formation but also transport PDGF-B, which acts synergistically with VEGF to stabilize the newly formed vessels [15]. Through the PDGF-B/PDGFRβ axis, platelets recruit pericytes and smooth muscle cells, promoting rapid vascular maturation [16]. This phenomenon is particularly notable in clear cell carcinomas and is associated with poorer clinical outcomes [17].

In addition to releasing pro-angiogenic factors, platelets can also modulate the TME in other ways to promote angiogenesis. They interact with endothelial cells, pericytes, fibroblasts, and immune cells in the TME [13,14], and they release cytokines and chemokines, creating favorable conditions for angiogenesis and tumor progression [13,14]. In addition, platelets can recruit and activate tumor-associated macrophages (TAMs), which also promote angiogenesis and tumor progression. TAMs release additional pro-angiogenic factors, such as matrix metalloproteinases (MMPs) and cytokines, further enhancing angiogenesis [18,19].

### 1.3. Platelets and Immune Evasion in Cancer

One of the key challenges in cancer therapy is the ability of tumor cells to evade the host immune system. Cancer cells have evolved numerous strategies to avoid immune detection and destruction, and platelets play a crucial role in protecting tumor cells from immune system attacks. Though not classified as immune system cells, platelets are extensively involved in the functions of several immune cells, including natural killer (NK) cells, monocytes, and certain T lymphocytes. Furthermore, the cytokines and other mediators released by platelets can suppress immune responses [20,21]. In addition, platelets can form aggregates with tumor cells, protecting them from shear stress and immune surveillance in the bloodstream [21,22]. This phenomenon, known as “platelet cloaking”, shields circulating tumor cells (CTCs) from NK cell-mediated cytotoxicity, thereby enhancing their survival and metastatic potential. This cloaking is mediated by the expression of “self” antigens on platelets, such as major histocompatibility complex (MHC) class I molecules, which mask the tumor cells and prevent their recognition by NK cells [21,23], a process that is crucial for protection against the immune system [24]. Moreover, platelets promote the expression of immune checkpoint molecules such as programmed death ligand 1 (PD-L1) on tumor cells, which also facilitates immune evasion [21,25].

Beyond platelet cloaking, platelets actively suppress T-cell responses and undermine T-cell-based immunotherapies. They impair these treatments—such as bispecific antibodies targeting PSMA—by reducing the recruitment and effector functions of CD4+ and CD8+ T cells. Platelet activation through TGF-β signaling leads to diminished T-cell degranulation, perforin release, and target cell lysis, ultimately compromising therapeutic efficacy. Targeting platelet function or inhibiting the TGF-β pathway may help restore T-cell reactivity and enhance treatment outcomes [26].

Platelets play a multifaceted role in modulating immune cell behavior within the TME, contributing to an immunosuppressive state that supports tumor progression. They release immunosuppressive cytokines like TGF-β, which inhibits the activation and function of cytotoxic T cells and NK cells, thereby impairing the immune response against tumor cells. Additionally, platelet-derived TGF-β can convert effector T cells into regulatory T cells (Tregs), which further suppress the immune response and promote tumor tolerance [21,27]. In this context, the TGFβ-docking receptor glycoprotein A repetitions-predominant (GARP)–TGF-β axis plays a pivotal role in tumor immune evasion by activating latent TGF-β on platelets and Tregs, thus promoting immune suppression and tumor progression [28,29,30]. Platelets also influence monocyte differentiation, inducing the formation of myeloid-derived suppressor cells (MDSCs), which inhibit T-cell responses and drive tumor growth [31].

In bladder cancer, platelets interact with stromal components of the TME, such as macrophages and fibroblasts, to promote an immunosuppressive microenvironment. For instance, they enhance the recruitment of M2 macrophages, which suppress antitumor immunity and contribute to tumor aggressiveness [32].

Moreover, platelets can promote the differentiation of monocytes into TAMs with a pro-tumorigenic M2 phenotype, enhancing the immune evasion capability of the tumor [21]. In summary, by shaping immune cell behavior, platelets play a critical role in facilitating immune evasion and supporting tumor growth and metastasis.

### 1.4. Platelets in Cancer Metastasis

Metastasis is the process by which cancer cells spread from the primary tumor to other sites in the body. This complex, multistep process involves the detachment of tumor cells from the primary site, invasion into surrounding tissues, intravasation into the bloodstream, survival within the circulatory system, extravasation at a distant location, and colonization of a new organ [33]. When tumor cells enter the bloodstream, they encounter a hostile environment characterized by shear stress and immune surveillance. Platelets play a protective role through platelet cloaking, which not only shields the tumor cells from these hostile forces but also enhances the adhesive properties of CTCs, thus facilitating their interaction with the vascular endothelium and promoting extravasation into distant tissues [9,22,23,34].

This extravasation is a critical step in metastasis [33], and platelets play a pivotal role in facilitating this process by interacting with endothelial cells and promoting the formation of gaps in the endothelial barrier, thereby enabling tumor cells to exit the bloodstream [35]. To achieve this vascular permeability, platelets release factors such as ADP, thromboxane A2, and ATP, which activate endothelial cells, thus supporting the metastatic dissemination of tumor cells [7]. Moreover, platelets secrete lysophosphatidic acid, which enhances tumor cell adhesion to endothelial cells and promotes their transendothelial migration [36].

For example, in prostate cancer, platelet-derived lysophosphatidic acid (LPA) and CCL3L1 enhance tumor cell adhesion and survival via receptors such as LPAR3 and CCR1, which are notably overexpressed in aggressive disease. Interactions between platelets and prostate cancer cells through ephrin–EPH receptor signaling promote apoptotic resistance, while platelet-derived microvesicles transfer pro-invasive factors to tumor cells, further amplifying their metastatic potential. These mechanisms illustrate how platelets actively contribute to the metastatic cascade by recruiting pro-metastatic cells and facilitating bidirectional signaling that enhances tumor cell survival and invasion [37,38].

Additionally, platelets release matrix MMPs and thrombospondin-1 (TSP-1), which play critical roles in remodeling the extracellular matrix (ECM) and degrading the basement membrane. MMPs, including MMP-2 and MMP-9, are proteolytic enzymes that cleave key ECM proteins like collagen and fibronectin, creating pathways for tumor cells to invade surrounding tissues and extravasate into distant sites [39]. TSP-1, a multifunctional glycoprotein, not only activates MMP-2 but also interacts with integrins and other ECM components to facilitate cell migration [40,41].

Once tumor cells have extravasated, they need to survive and establish a metastatic niche to colonize the new organ. Platelets contribute to this step in the metastatic process by secreting factors that promote tumor cell survival and dormancy. For example, platelets release TGF-β and other cytokines that inhibit apoptosis and induce a quiescent state in tumor cells, allowing them to evade immune detection and survive in the new microenvironment. Moreover, TGF-β signaling promotes epithelial–mesenchymal transition (EMT) in tumor cells, facilitating their migration and invasion [7]. Furthermore, platelets can create a pre-metastatic niche by interacting with stromal cells and immune cells at distant sites, creating a favorable environment for tumor cell colonization. This includes recruiting bone marrow-derived cells, such as MDSCs and TAMs, which promote tumor cell survival and growth [18].

Taken together, through their multifaceted roles at different stages of the metastatic process—protecting tumor cells, promoting endothelial permeability, and shaping the metastatic niche—platelets serve as critical enablers of metastasis, enhancing the survival, dissemination, and colonization of cancer cells.

## 2. Tumor-Educated Platelets

In addition to the crucial role of platelets in several stages of cancer progression, tumor-educated platelets (TEPs) have shown great potential as biomarkers for the early detection of cancer, disease monitoring, and development of tailored therapeutic strategies, particularly in solid tumors. Here, we review the emerging potential of TEPs with a specific emphasis on urological malignancies.

TEPs constitute a distinct subpopulation of platelets that have been exposed to tumors. This exposure modifies the RNA and protein profiles of the platelets to reflect the molecular characteristics of the tumor [42]. This process, known as “tumor education”, enables platelets to support tumor growth, metastasis, and immune evasion more effectively [43]. This transformation is driven by direct interactions between tumor cells and platelets through adhesion molecules such as P-selectin, integrins, and glycoproteins. Additionally, it involves the activation of platelet receptors by tumor-released extracellular molecules, including thrombin, tissue factors, and extracellular vesicles (EVs). These EVs transfer tumor-derived proteins, lipids, and RNA to the platelets, enabling them to sequester spliced or unspliced mRNA and preserve transcripts from megakaryocytes, thereby altering the platelet transcriptome and behavior. TEPs exhibit significant transcriptomic changes, including the upregulation of PDGF and TGF-β1, which promote angiogenesis and tumor progression. Certain cancer-associated cytokines, such as granulocyte colony stimulating factor (G-CSF), granulocyte macrophage-colony stimulating factor (GM-CSF), and interleukin-6 (IL-6), enhance megakaryopoiesis and increase platelet counts, inducing tumor-educated megakaryocytes that play a pivotal role in the formation of TEPs, particularly under the influence of IL-6. With these adaptations, platelets acquire a protumorigenic phenotype, which enhances their ability to support angiogenesis, tumor invasion, and TME remodeling [44,45] (Figure 2).

In addition to their enhanced role as active mediators of tumor progression, TEPs have shown promise as valuable biomarkers detectable by minimally invasive liquid biopsies. Here, we focus on the analysis of TEPs as a promising tool for cancer detection, monitoring, prognosis, and personalization of treatment [46,47].

### 2.1. TEPs as Blood-Sourced Biomarkers for Solid Tumors

Platelets cannot synthesize RNA, and instead, their RNA is either endocytosed from blood circulation or derived from megakaryocytes. Tumor-derived signals can induce specific changes in platelets, and the RNA content of TEPs is dynamically regulated by the TME. This “tumor education” allows platelets to internalize tumor-derived biomolecules, including RNA, which confers an RNA profile in the platelets that reflects that of the tumor [44,48] (Figure 2). Interestingly, these changes can be detected and analyzed using high-throughput sequencing techniques [49]. This ability of TEPs to carry tumor-specific information positions them as a promising biomarker that can easily be analyzed in liquid biopsies of patients with solid tumors. Unlike traditional biopsies, which require invasive tissue sampling, liquid biopsies offer a less invasive alternative by utilizing easily accessible body fluids like blood. Moreover, the advantages of TEPs compared to other blood-based biosources—such as CTCs, exosomes, circulating tumor DNA (ctDNA), and cell-free nucleic acids—include their abundance, ease of isolation, stability and capacity to process RNA [50,51]. The analysis of TEPs in liquid biopsies thus provides a non-invasive method for the detection of cancer, monitoring of treatment response, and identification of potential therapeutic targets [42,46,47,48].

In 2010, Calverley and colleagues were the first to analyze platelet RNA in patients with solid tumors. Using microarray analysis, they profiled platelet mRNA from seven healthy individuals and five patients with untreated metastatic non-small cell lung cancer (NSCLC). Their study identified 200 altered RNAs between the two groups, with 197 transcripts showing decreased levels in the platelets of NSCLC patients. This pilot study provided the first evidence that platelet RNA could serve as a potential diagnostic tool [52].

Nilsson and colleagues later observed that tumor-derived RNA can be transferred into blood platelets and that platelets from prostate and glioma cancer patients contain cancer-associated RNA biomarkers [53]. In 2016, the same group employed RT-PCR to analyze echinoderm microtubule associated protein-like 4 (EML4)-anaplastic lymphoma kinase (ALK) rearrangements in platelets and plasma from 77 NSCLC patients. They observed a 65% sensitivity and 100% specificity for detection. Moreover, in a subgroup of patients with EML4-ALK-rearranged tumors undergoing targeted therapy, the authors tracked changes in EML4-ALK-rearranged RNA levels in platelets over the course of treatment and found that these levels correlated with progression-free survival (PFS) and overall survival (OS). Notably, monitoring platelet EML4-ALK expression predicted resistance to targeted therapy approximately two months before conventional imaging methods detected disease progression [54].

In a salient study by Best and colleagues, mRNA sequencing of TEPs from 283 samples was able to distinguish 228 patients with cancer from 55 healthy individuals with 96% accuracy. Moreover, among six different tumor types (NSCLC, colorectal cancer, glioblastoma, pancreatic cancer, hepatobiliary cancer, and breast cancer), the location of the primary tumor was correctly identified with 71% precision. Furthermore, the study identified several genetic alterations, including MET and ERBB2 positivity and mutations in Kirsten rat sarcoma virus (KRAS), epidermal growth factor receptor (EGFR), phosphatidylinositol-4,5-bisphosphate 3-kinase, and catalytic subunit alpha (PIK3CA) [42].

Along the same lines, Sol et al. demonstrated that TEP-derived RNA signatures, identified using swarm intelligence, were able to detect and monitor glioblastoma. The RNA profiles of TEPs from glioblastoma patients were distinguishable from those of patients with metastatic brain cancer, multiple sclerosis, and healthy controls, with an accuracy exceeding 80% in all cases. Additionally, the study provided evidence that tumor signals in TEPs are dynamic, which enabled pseudoprogression to be differentiated from true progression [55].

Building on the foundational work of these studies, which pioneered the exploration of TEPs and demonstrated their potential as diagnostic, prognostic, monitoring, and therapeutic targeting tools, subsequent research has further emphasized the utility of TEPs across various tumor types, including sarcoma, nasopharyngeal carcinoma, colorectal cancer, and NSCLC. Heinhuis and colleagues focused on the analysis of TEPs in sarcoma. Using RNA-seq, they analyzed TEP samples from patients with various sarcoma subtypes as well as from control groups of former sarcoma patients in remission and from healthy donors. They found significant transcriptional changes in TEPs that were able to distinguish between affected patients and controls with an accuracy exceeding 85%. Additionally, their gene expression analysis suggested that TEPs can differentiate between different sarcoma subtypes. These findings highlight the potential of analyzing TEP RNA in liquid biopsy as a diagnostic tool for sarcoma [56]. In a study of nasopharyngeal carcinoma, Wang et al. identified two TEP microRNAs (miR-34c-3p and miR-18a-5p) that were upregulated in patients compared to healthy donors, suggesting their promise as biomarkers for disease detection. Notably, these expression changes were absent in plasma, highlighting a “tumor education” effect occurring specifically within the platelets [57]. In a study of colorectal cancer, Yang et al. found that TIMP metallopeptidase inhibitor 1 (*TIMP1*) mRNA levels were elevated in platelets from patients compared to healthy individuals or those with inflammatory bowel diseases, indicating a potential as a diagnostic biomarker for colorectal cancer [58]. Most recently, in advanced NSCLC, Hu and colleagues evaluated PD-L1 expression levels in tumor tissue and TEPs in patients treated with immunotherapy. Although expression levels did not correlate between tumor tissue and TEPs, elevated PD-L1 mRNA levels in TEPs were associated with improved outcomes, underscoring their potential as a surrogate biomarker to guide treatment decisions [59].

Taken together, the findings from these studies strongly support the potential of TEPs as a blood-sourced biomarker for detecting and monitoring various tumors. TEPs offer unique advantages, such as their dynamic RNA content, which reflects real-time tumor activity and enables the identification of cancer-specific signatures. In the following sections, we focus on the potential role of TEPs as prognostic and predictive biomarkers in prostate, kidney, and bladder cancers. Understanding the specific alterations in TEP RNA profiles within these urological malignancies could pave the way for improved patient stratification, non-invasive disease monitoring, and enhanced therapeutic decision-making.

Table 1 provides a summary of the main studies conducted on TEPs in prostate, kidney, and bladder cancers. It highlights key findings, methodologies, and clinical implications and offers a comprehensive overview of the current state of research in these urological malignancies.

### 2.2. TEPs and Prostate Cancer

Prostate cancer is one of the most common malignancies in men, second only to lung cancer [65]. Most cases are detected at a localized stage, requiring transrectal or transperineal biopsy, which are invasive procedures with potential complications. In advanced cases, metastases frequently spread to the bones and retroperitoneal lymph nodes, making it difficult to obtain biopsy samples [66]. In fact, bone biopsies are particularly challenging, as the decalcification process required for sample preparation can degrade nucleic acids, limiting the ability to perform molecular analyses [67]. Additionally, tissue biopsies often fail to reflect tumor heterogeneity. Given these limitations, liquid biopsy has emerged as a minimally invasive and highly promising alternative for molecular profiling in prostate cancer [68]. Among the various biosources available, TEPs stand out for their ability to dynamically reflect the molecular characteristics of the tumor.

The first findings on TEPs in prostate cancer were reported in 2011 by Nilsson and colleagues. They identified the presence of EGFR variant III (EGFRvIII) and prostate cancer antigen 3 (PCA3) in platelets isolated from glioma and prostate cancer patients, respectively. Moreover, significant differences in RNA profiles were observed in platelets from glioma patients compared to those of healthy controls, suggesting a distinct cancer signature [53].

Building on Nilsson’s findings, Hänze and colleagues conducted a pilot study to evaluate whether TEPs could be used for early detection of prostate cancer. Using relative quantitative RT-PCR, they analyzed RNA in platelets from 31 patients with localized prostate cancer undergoing radical prostatectomy and 29 healthy young men as controls. The study included a variety of RNA markers, such as PCA3 and its isoforms, metastasis-associated lung adenocarcinoma transcript 1 (MALAT1), enhancer of zeste homolog 2 (EZH2), prostate-specific antigen (PSA), and prostate-specific membrane antigen (PSMA). The results revealed heterogeneous expression patterns of RNA markers in platelets, with no statistically significant differences between the groups. PSA was detected in only two patients (10%) and was absent in all control samples, but this difference was not statistically significant. The authors concluded that while TEPs were not effective for early detection in this study, certain RNA markers in platelets, such as PCA3 and MALAT1, may still hold potential as biomarkers for prostate cancer. They emphasized the need for larger, prospective studies to further investigate the role of platelets in prostate cancer diagnosis and to determine whether TEPs can ultimately serve as reliable biomarkers [60].

In this context, In ’t Veld and colleagues developed and validated a pan-cancer detection test using platelet RNA analysis, introducing a promising method for cancer detection and localization. Their study utilized a dataset of 2351 samples, which included 18 tumor types, asymptomatic controls, and symptomatic controls without a cancer diagnosis. Their pan-cancer ThromboSeq algorithm was able to distinguish between 18 tumor types with high specificity. Notably, prostate cancer was the most frequently detected tumor type, identified in 92% of cases, with an impressive area under the curve (AUC) of 0.98. Moreover, the accuracy of detection varied with tumor stage, showing improved performance in advanced stages, ranging from 46% for stage I to 72% for stage IV. Furthermore, the study accurately identified the tumor origin in over 80% of cancer patients for five specific tumor types, including a combined category encompassing urothelial, prostate, and renal tumors [61].

For patients with prostate cancer who experience disease relapse after local therapy or who have metastatic disease, androgen deprivation therapy (ADT) remains the cornerstone of systemic treatment. However, despite significant initial responses to ADT, nearly all metastatic patients eventually progress to incurable metastatic castration-resistant prostate cancer (mCRPC) [69]. Using digital PCR, Tjon-Kon-Fat and colleagues analyzed TEPs to identify cancer-derived transcripts in mCRPC patients before initiating therapy. The study included 24 patients receiving docetaxel-based chemotherapy as first-line treatment; 26 patients treated at any stage of castration-resistant disease with abiraterone, an irreversible inhibitor of the cytochrome P450 17α-hydroxy/17,20-lyase (CYP17) enzyme responsible for androgen synthesis; and 15 healthy controls. Transcripts of prostate cancer-associated biomarkers—kallikrein-related peptidase-2 and -3 (KLK2, KLK3), folate hydrolase 1 (FOLH1), and neuropeptide Y (NPY)—were detected in the platelets of cancer patients but not in those of healthy controls. In the docetaxel-treated cohort, no biomarkers correlated with PFS, and only detectable FOLH1 levels were linked to shorter OS (*p* < 0.001). However, in the abiraterone-treated cohort, FOLH1 was associated with a poor PSA response (*p* < 0.05), while patients with detectable levels of KLK3, KLK2, and FOLH1 exhibited significantly increased risks of death, with hazard ratios of 4.7 (*p* < 0.01; 95% CI, 1.6–13.6) for KLK3, 5.3 (*p* < 0.001; 95% CI, 1.9–14.5) for KLK2, and 3.0 (*p* < 0.05; 95% CI, 1.1–8.1) for FOLH1. Moreover, KLK3 (HR 2.6, *p* < 0.01; 95% CI, 1.1–6.2), NPY (HR 2.7, *p* < 0.05; 95% CI, 1.0–6.8), and FOLH1 (HR 3.4, *p* < 0.05; 95% CI, 1.2–9.8) were all associated with shorter PFS. Notably, a three-gene panel comprising KLK3, FOLH1, and NPY effectively distinguished responders from non-responders to abiraterone therapy, with patients positive for all three biomarkers showing a 4.2-fold increased risk of therapy failure (*p* < 0.01) and the panel achieving 87% sensitivity and 82% specificity for identifying patients with PFS longer than 6.5 months, offering a robust tool to identify resistance to treatment [59].

In summary, TEP analysis in liquid biopsies shows promise as a non-invasive tool for molecular profiling and treatment monitoring in prostate cancer. While early detection remains challenging, the ThromboSeq algorithm and biomarker panels highlight the potential of TEPs in identifying tumor characteristics, predicting therapeutic outcomes, and enhancing personalized treatment strategies (Table 1).

### 2.3. TEPs and Kidney Cancer

Kidney cancer ranks as the 14th most common cancer globally across both sexes. By 2045, the incidence of kidney cancer is projected to reach 616,030 cases with an estimated 256,427 deaths [65]. RCC accounts for approximately 85% of kidney tumors, with clear cell histology comprising around 70% of RCC cases. The diagnostic gold standard remains histologic evaluation via surgical resection or biopsy. While 70% of RCC cases are diagnosed at a localized stage, the remaining 30% present with metastatic disease. Despite curative-intent treatments such as surgery or focal therapies, one-third of localized tumors recur during follow-up, significantly increasing the risk of mortality, although survival rates have improved, with localized RCC achieving a 5-year survival rate of 90% [70,71].

Although early detection is critical to further improving survival outcomes in RCC, unlike prostate cancer, where the PSA test is part of a standard population screening program, there is currently no equivalent screening method for RCC. Early detection thus remains a significant challenge, largely due to the low clinical manifestation of symptoms in localized stages, resulting in most diagnoses being incidental findings during radiological tests conducted for unrelated reasons. In recent years, there has been growing interest in the use of blood biomarkers and liquid biopsy for the diagnosis and management of RCC [72,73]. While TEPs remain among the least explored biomarkers in this context, the established prognostic value of platelets—whose count is already integrated into risk stratification models for metastatic RCC treated with VEGF-targeted agents [74] and is correlated with aggressive tumor features in localized RCC [75]—underscores the ongoing need for more extensive research in this field.

In this evolving landscape, Kidney Injury Molecule-1 (KIM-1) has emerged as a particularly promising candidate. It is a transmembrane glycoprotein encoded by the *HAVCR1* gene, notably overexpressed in clear cell and papillary RCC. Its soluble ectodomain, detectable in both plasma and serum, enhances malignancy prediction when measured preoperatively and serves as an indicator of minimal residual disease, recurrence risk following nephrectomy, and potential benefit from adjuvant immunotherapy, as highlighted in recent studies [76,77,78]. In 2020, Sol and colleagues employed RNA panels derived from TEPs, identified through the digital SWARM algorithm, which leverages swarm intelligence, to detect and monitor glioblastoma. Their study assessed the accuracy of detection by comparing spliced RNA profiles from glioblastoma patients with those of individuals diagnosed with multiple sclerosis or brain metastases and healthy controls. Intriguingly, among the patients with brain metastases, seven were identified as having RCC. However, the study did not include a detailed subgroup analysis specific to the RCC cohort, leaving a gap in understanding the applicability of this approach to RCC-related metastases [55].

In a study more specifically focused on RCC, Xiao and colleagues investigated the use of TEPs as a minimally invasive biomarker for detecting RCC. Using RNA sequencing (RNA-seq), they analyzed platelet RNA profiles from 24 RCC patients and 25 controls (including healthy donors and individuals with benign renal tumors) and identified 203 differentially expressed genes, with 64 upregulated and 139 downregulated in RCC patients. With machine learning techniques, they developed a 68-gene TEP-based diagnostic model that showed high accuracy across datasets: 100% (AUC: 1.000) in the training set, 88.9% (AUC: 0.963) in the validation set, and 95.9% (AUC: 0.988) in the overall cohort. Despite certain limitations, including the small sample size and lack of external validation, these findings underscore the potential of TEPs as a robust and accurate blood-based biomarker platform, providing a non-invasive alternative for RCC detection and paving the way for further validation and clinical application [63].

In the study by In ’t Veld and colleagues using ThromboSeq [61], the test achieved an AUC of 0.87 and a prediction accuracy of 66% in the sub-cohort of RCC patients. Furthermore, the test successfully identified the tumor origin in over 80% of cancer patients, including a category grouping urothelial, prostate, and renal tumors. Platelet RNA profiles were shown to be influenced by both primary tumors and metastases, particularly brain metastases. Nevertheless, although RCC was among the tumor types analyzed, detailed results specific to this subgroup were not reported [61].

In conclusion, TEP RNA analysis shows great potential as a non-invasive tool for RCC diagnosis, offering high accuracy for early detection and tumor characterization. However, its clinical use is limited by studies with small sample sizes, lack of validation, and incomplete mechanistic understanding (Table 1). Large-scale studies are needed to validate findings, optimize diagnostic models, and explore the role of TEPs in monitoring metastasis and treatment response, thereby paving the way for enhanced RCC management and broader applications.

### 2.4. TEPs and Bladder Cancer

Urinary bladder tumors are the 9th most common cancer overall and the 6th among men. Projections indicate a substantial rise in the incidence of bladder cancer over the next two decades, with cases potentially surpassing one million [65,79]. More than 90% of urothelial tumors originate in the urinary bladder, with urothelial carcinoma being the predominant histological subtype. It is essential to differentiate between non-muscle-invasive and muscle-invasive urothelial tumors, as their prognosis and management differ significantly. Approximately 75% of cases are non-muscle-invasive, which typically has a lower risk of recurrence and progression. As a result, aggressive therapeutic approaches are generally unnecessary for these patients. Instead, treatment focuses primarily on endoscopic interventions aimed at minimizing the risks of recurrence and progression [80,81].

Muscle-invasive bladder cancer (MIBC) has a global 5-year mortality rate of approximately 40–50% and poses a significant clinical challenge due to its aggressive biological behavior. Treatment decisions for MIBC are predominantly based on histopathological evaluation of biopsies from primary tumors and clinical staging, which together guide the therapeutic approach [82]. The standard treatment for patients with MIBC involves neoadjuvant cisplatin-based chemotherapy followed by radical cystectomy, which offers the best chance for a curative outcome. Recently, combining chemotherapy with immunotherapy has also demonstrated improved outcomes [83]. Nevertheless, despite this aggressive multimodal approach, metastatic recurrence remains frequent, and the 5-year survival rate is limited to 50–60% [84,85]. These suboptimal outcomes underscore the urgent need for reliable prognostic and predictive biomarkers to refine treatment strategies. Such biomarkers could help identify patients who may be able to avoid invasive procedures like cystectomy, thereby preserving their quality of life without compromising outcomes. Current research is actively evaluating novel blood-based biomarkers, such as CTCs, ctDNA, urine proteins, exosomes, and metabolites, which show potential for advancing personalized medicine in the management of MIBC [86,87].

The use of ctDNA as a predictive and prognostic biomarker represents one of the most advanced and clinically applicable approaches in this disease [88,89,90,91]. For example, the Phase 3 IMvigor010 trial evaluated the efficacy of one year of immune checkpoint inhibitor therapy with adjuvant atezolizumab, a PD-L1 inhibitor, following cystectomy in patients with high-risk MIBC. Although 809 patients were randomized, the study did not meet its primary endpoint of improving disease-free survival [92]. However, this trial also explored the role of ctDNA as a biomarker for predicting outcomes and guiding treatment decisions. Patients with detectable ctDNA had worse survival outcomes although they also derived benefit from atezolizumab, with significantly higher survival rates and evidence of ctDNA clearance or reduction [93,94]. Along the same lines, results from the KEYNOTE-361 study, presented at the 2024 ASCO Annual Meeting, showed that ctDNA levels act as a prognostic indicator for patients with advanced urothelial carcinoma receiving pembrolizumab, particularly in evaluating treatment response as compared to traditional chemotherapy [95]. These findings underscore the potential of ctDNA positivity and clearance as key indicators of clinical outcomes, supporting the adoption of a personalized approach for patient selection in postoperative cancer treatment.

Unlike ctDNA, the role of TEPs in bladder cancer has not been extensively investigated. However, Hinsenveld and colleagues designed a prospective study aimed at evaluating the potential of predicting a complete pathological response in patients with MIBC undergoing neoadjuvant chemotherapy. The ultimate goal is to identify patients who may safely avoid radical cystectomy. Their approach involves integrating clinical–radiological, histological, and molecular biomarker parameters derived from tissue, urine, and blood samples, including the exploration of TEPs as part of this multimodal assessment [64].

In the study by In ’t Veld and colleagues, the pan-cancer ThromboSeq assay achieved an 89% accuracy rate and an AUC of 0.99 in patients with urothelial cancer and accurately detected the tumor origin in over 80% of cases within a mixed cohort [61].

In conclusion, while emerging biomarkers such as ctDNA have shown substantial promise in predicting prognosis and guiding treatment strategies in bladder cancer, the potential role of TEPs remains largely unexplored in this context, highlighting the need for more comprehensive studies to evaluate the usefulness of TEPs as predictive and prognostic biomarkers (Table 1).

## 3. Conclusions and Future Perspectives

TEPs are a promising source of tumor molecular data. They are easily accessible by liquid biopsy and offer unique advantages due to their abundance, ease of isolation, and dynamic RNA profiles that reflect tumor activity in real time. Current evidence underscores their potential as diagnostic, prognostic, and predictive biomarkers across a variety of solid tumors, including urological cancers. Unlike traditional tumor tissue biopsies, which are invasive and often fail to capture tumor heterogeneity, TEPs in liquid biopsies provide a minimally invasive alternative capable of delivering comprehensive molecular insights that can guide tailored treatment strategies to improve patient outcomes.

TEPs offer several distinct advantages over other blood-based biosources like ctDNA. While ctDNA can be fragmented and is often present at low concentrations, particularly in early-stage cancers or certain tumor types, TEPs are abundant and can be isolated using standardized techniques. Their ability to internalize RNA and other tumor-derived molecules provides a dynamic profile of the tumor molecular landscape. Moreover, they capture a broader range of signals, including RNA transcripts and splicing variants, which may not be detectable by ctDNA analysis. Importantly, TEPs reflect both primary and metastatic tumor activity, making them a versatile tool for cancer monitoring and guiding treatment decisions. Nonetheless, several technical challenges remain before TEP analysis can be widely adopted in clinical practice. Reproducibility is a major concern, and the lack of standardized protocols for platelet isolation, RNA extraction, and downstream bioinformatic analysis may lead to inconsistent results across studies. Although the team led by Thomas Wurdinger has proposed a robust and reproducible method for platelet isolation from whole blood for sequencing-based studies [42,96], its widespread adoption and validation across independent cohorts and clinical settings is still pending. Another key challenge lies in minimizing molecular noise, which may arise from platelet interactions with non-cancerous cells or systemic inflammatory processes. One strategy to address this issue involves parallel analysis of gene expression in tumor tissue to validate that the biomarkers identified in TEPs truly originate from the tumor and are not confounded by other physio pathological conditions. This comparative approach has been employed in several studies [53,60], reinforcing the tumor-specific nature of selected TEP-derived RNA signatures. Addressing these methodological limitations is critical for enhancing the reliability, sensitivity, and specificity of TEP-based assays, ultimately paving the way for their integration into clinical decision-making workflows.

In urological tumors, the role of TEPs is particularly compelling though underexplored. In prostate cancer, emerging evidence highlights the potential of TEP RNA profiles to detect early disease and predict therapeutic responses, although further validation in prospective, multicenter studies is required. In kidney cancer, preliminary studies have demonstrated high diagnostic accuracy, but the clinical relevance of these findings needs to be evaluated in larger cohorts. Notably, given the current challenges in early detection and the lack of effective screening strategies for RCC, TEP analysis could represent a valuable non-invasive diagnostic approach, potentially enabling earlier intervention and improved outcomes. In bladder cancer, the exploration of TEPs remains limited but with a potential application for monitoring treatment response and guiding bladder preservation strategies. Future research should focus on integrating TEP analysis into multimodal biomarker panels to enhance its predictive power, particularly in MIBC patients undergoing neoadjuvant chemotherapy. This approach could reduce treatment delays, prevent toxicities in non-responsive patients, and optimize the timing of surgery. It is also worth noting that testicular germ cell tumors and penile cancer have not been included in this review, as—likely due to their low incidence—no studies have yet investigated the potential prognostic or predictive role of TEPs in these malignancies. This represents a gap in current knowledge that deserves to be explored in future research.

Despite the promising potential of TEPs, several important questions remain unanswered. One key area is the longitudinal stability of TEP RNA signatures over the course of disease progression and treatment—understanding whether these profiles remain consistent or evolve in response to therapeutic pressure could enhance their value in monitoring. Additionally, it is still unclear to what extent TEPs are able to capture intratumoral heterogeneity or reflect emerging resistance mechanisms in real time. Further research is also needed to determine how TEP profiles vary among different subtypes and stages of urological cancers and whether they can be tailored to specific clinical scenarios. Finally, the cost effectiveness, scalability, and integration of TEP-based assays into existing diagnostic and therapeutic workflows remain open challenges that must be addressed before widespread clinical implementation can be achieved.

In conclusion, while significant progress has been made, the clinical integration of TEPs as blood-based biomarkers requires robust validation and technological refinement. Expanding research efforts in this field will be crucial to unlocking the full potential of TEPs and transforming their promise into tangible benefits for patients, particularly in the realm of urological tumors.

## Figures and Tables

**Figure 1 ijms-26-03595-f001:**
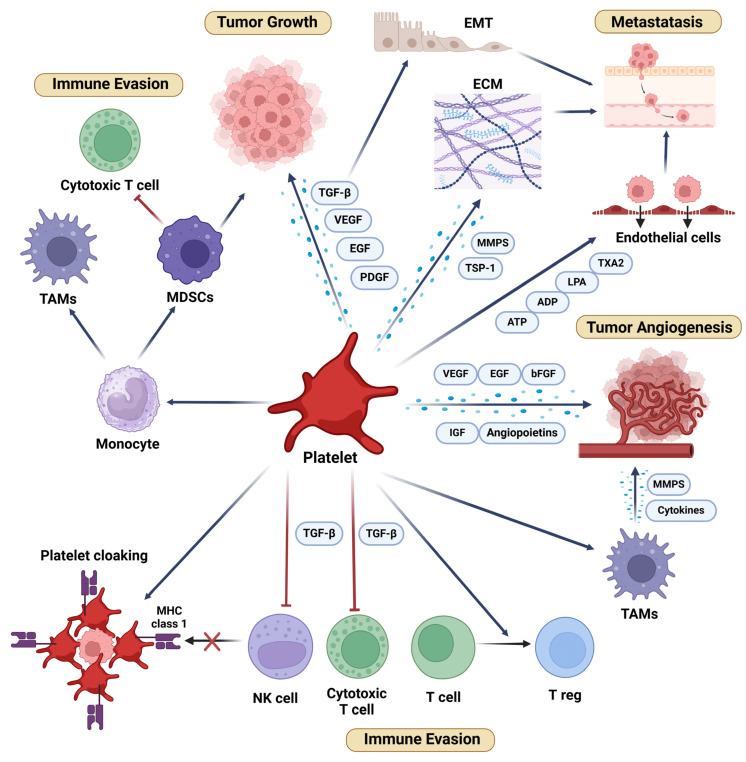
Platelets significantly influence various aspects of cancer development, including tumor proliferation, angiogenesis, immune evasion, and metastasis. These processes are mediated through direct or indirect interactions with tumor cells and the tumor microenvironment, involving the secretion of growth factors, chemokines, and the modulation of immune cells, which play pivotal roles in shaping the tumor microenvironment and driving cancer progression. bFGF: Basic Fibroblast Growth Factor; ECM: Extracellular Matrix; EGF: Epidermal Growth Factor; EMT: Epithelial–Mesenchymal Transition; IGF: Insulin Growth Factor; LMA: Lysophosphatidic Acid MDSCs: Myeloid-Derived Suppressor Cells; MHC: Major Histocompatibility Complex; MMPs: Metalloproteinases; NK Cell: Natural Killer Cell; PDGF: Platelet-Derived Growth Factor; TAMs: Tumor-Associated Macrophages; TGF-β: Transforming Growth Factor-Beta; Treg: Regulatory T Cell; TSP-1: Thrombospondin-1; TXA2: Thromboxane A2; VEGF: Vascular Endothelial Growth Factor. Ͱ: Inhibition; ←: Activation. Created in https://BioRender.com (accessed on 7 January 2025).

**Figure 2 ijms-26-03595-f002:**
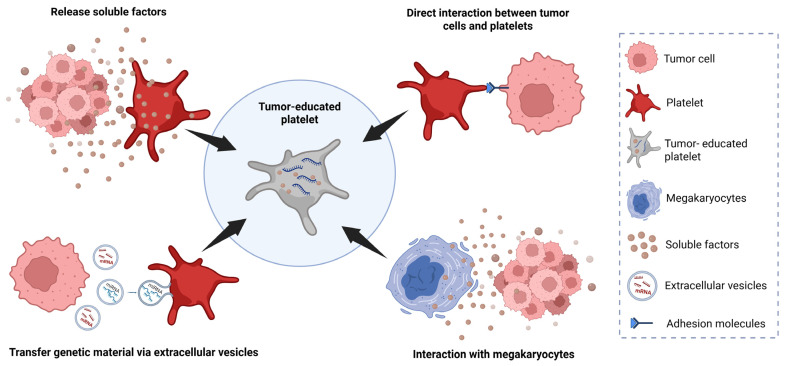
Mechanisms of “tumor education” of platelets. These include the secretion of soluble tumor-derived factors, direct interactions between platelets and tumor cells, the transfer of genetic material via extracellular vesicles, and the interaction with megakaryocytes. Created in https://BioRender.com (accessed on 10 April 2025).

**Table 1 ijms-26-03595-t001:** Summary of key studies investigating tumor-educated platelets (TEPs) in prostate, kidney, and bladder cancers.

Prostate Cancer						
Study Reference	Objective of TEP Analysis	Gene Expression Signature	Method of Analysis	Nº Cases/Controls	Key Findings	Clinical Implications
[53]	Detection of tumoral RNA in platelets	*PCA3*	RT-PCR	12 patients/10 controls	PCA3 RNA detected in prostate cancer patients	Potential for prostate cancer detection
[60]	Early tumor detection	*PCA3*,* MALAT1*,* EZH2*,* AMACR PSGR*,* PSA*,* PSMA*,* TRPM8*	*RT-PCR*	31 patients/29 controls	No significant differences between cases and controls	TEPs may not be suitable for early tumor detection
[61]	Tumor detection and tumor origin identification	Broad pan-cancer RNA	ThromboSeq (RNA-seq)	35 patients	Detected tumor in 92% (AUC = 0.98), improved with higher stage; origin identified in >80%	High potential for detecting advanced prostate cancer
[62]	Prediction of response to therapy	*KLK2*,* KLK3*,* FOLH1*,* NPY*	Digital PCR	50 patients/15 controls	A three-gene panel (KLK3, NPY, FOLH1) identified resistance to abiraterone therapy	Predictive/prognostic tool for mCRPC patients
**Kidney Cancer**						
**Study** **Reference**	**Objective of TEP Analysis**	**Gene Expression Signature**	**Method of Analysis**	**Nº Cases/Controls**	**Key Findings**	**Clinical Implications**
[63]	Diagnostic	68-gene panel	RNA-seq	24 patients/25 controls	Diagnostic accuracy of 95.9% (AUC: 0.988)	Significant potential for RCC blood-based screening
[61]	Tumor detection and tumor origin identification	Broad pan-cancer RNA	ThromboSeq (RNA-seq)	28 patients	Detection accuracy of 66% (AUC = 0.87); tumor origin identified in >80%	Demonstrated potential for liquid biopsy though limited renal-specific data
**Bladder Cancer**						
**Study** **Reference**	**Objective of TEP Analysis**	**Gene Expression Signature**	**Method of Analysis**	**Nº Cases/Controls**	**Key Findings**	**Clinical Implications**
[61]	Tumor detection and tumor origin identification	Broad pan-cancer RNA	ThromboSeq (RNA-seq)	28 patients	Detection accuracy of 89% (AUC 0.99); tumor origin identified in >80%	Demonstrated potential for liquid biopsy though limited bladder-specific data
[64]	Predict pathological response to neoadjuvant chemotherapy	Not specified	WES on tissue samples, followed by qPCR on liquid biopsy	150 patients	No results	Potential for bladder preservation by avoiding unnecessary cystectomies

**AUC:** area under the curve; **mCRPC**: metastatic castration-resistant prostate cancer; **RCC**: renal cell carcinoma; **TEPs**: tumor-educated platelets.

## Data Availability

Not applicable.

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
