# Peer review of "Tumor-Educated Platelets in Urological Tumors: A Novel Biosource in Liquid Biopsy"

_ijms, 2025, doi:10.3390/ijms26083595_

Round 1

Reviewer 1 Report

Comments and Suggestions for Authors

Starts with a great, in depth and well written review of platelets in cancer pathogenesis and progression. Given this is a review focused on GU malignancies, would suggest adding more to this section with regards to the diseases of interest. In the first several paragraphs and sections, it is wonderful to read, but I wonder if any of the data pertains to GU malignancies, or models in GU, vs. other tumor types? The reader has to specifically look up the citation to see if the background pertains to GU or for other cancers.

I would also suggest including other references, such as how platelet count has been a prognostic marker (for RCC) and part of risk stratification for that disease (and other tumors). Given the focus on growth factors, the authors might delve into the data regarding the safety and use of platelet-derived growth factor administration (and EPO) as this commonly comes up for treating oncologists, particularly in GU malignancies, and is relevant to the pathogenesis presented in the background.

Along those lines, the data presented regarding immune cloaking – is this performed in GU cancers? Does it contribute to the immunogenicity of some GU tumors vs. others, e.g., renal vs. urothelial vs. prostate, and the “cold microenvironment” seen in the latter?

The paper does not really become GU focused until page 9-10, thus would add in GU relevance in the background/ beginning as suggested if it’s truly a review for GU cancers.

In section 2.1, the authors mention “other blood biomarkers” vs. Teps. What other blood-based biomarkers are they referring to? Suggest listing outright rather than just a few citations and this blanket phrase. Although the purpose of this and the next few sections is to review TEPs, there are MANY blood-based biomarkers in the development of GU cancers, and many have recently presented at large conferences. Also, the sensitivity/specificity in comparison to PSA must be explored for the prostate, as this is the benchmark we compare all other biomarkers to in this disease. Although this section of the paper is novel, it is the main focus and feels too short, and like it is missing a lot of other relevance to the field by not even mentioning the emergence of other blood based biomarkers in various stages of development, and the role and context of TEP within that landscape.

The authors own self-citation from ESMO 2018 feels out of place particularly as many other more mainstream data is not included as stated

Section 2.2- citation 52; prostate is the second most common malignancy in men apart from cutaneous diseases, NOT lung cancer. Perhaps the authors are referring to mortality?

Section 2.3 RCC: per above, would mention platelets as early prognostic scores for this disease and tie back to biology, if able and to the tests the authors mention. Also, context of other markers (like KIM-1, which has been presented at international conferences and being used in clinical trials to guide therapy) should be mentioned.

Section 2.4- the authors do a good job of mentioning ctDNA and CTC in this space, and contextualizing the TEP landscape in this disease in that respect (relatively low uptake and development, as there are these other validated liquid biopsies being utilized)

Why is testis cancer not included? No data? there are microRNAs being evaluated in testis cancer and standard of care tumor markers that are blood-based biomarkers we routinely use; it feels like it should be included.

Reviewer 2 Report

Comments and Suggestions for Authors

The manuscript submitted for my review is a carefully prepared narrative review focusing on tumor-educated platelets in urological tumors. The authors address a highly relevant and timely clinical topic with significant diagnostic and translational potential.

The paper presents, in a clear and systematic manner, a step-by-step overview of the role of platelets in tumor growth and progression, tumor angiogenesis, immune evasion in cancer, and metastasis formation. Subsequent sections characterize the phenomenon of tumor-educated platelets as a potential source of blood-sourced biomarkers for solid tumors, offering promising perspectives for diagnostics, disease monitoring, and assessment of treatment susceptibility.

Particular recognition is due to the section discussing the possible clinical implications of the described mechanisms, as well as the well-reasoned and literature-based proposals for diagnostic applications. The manuscript is further enhanced by carefully constructed and visually clear illustrations.

Given the thoroughness of the review, its extensive references to current scientific literature, and the high quality of the presented content, I have no critical remarks or suggestions. I consider the manuscript acceptable in its current form.

Round 2

Reviewer 1 Report

Comments and Suggestions for Authors

great job addressing concerns and making this very informative and well rounded review